# Omega-3 Polyunsaturated Fatty Acids Counteract Inflammatory and Oxidative Damage of Non-Transformed Porcine Enterocytes

**DOI:** 10.3390/ani10060956

**Published:** 2020-05-31

**Authors:** Tamil Selvi Sundaram, Carlotta Giromini, Raffaella Rebucci, Antonella Baldi

**Affiliations:** 1Department of Veterinary Science for Health, Animal Production and Food Safety, University of Milan, Via Trentacoste 2, 20134 Milan, Italy; carlotta.giromini@unimi.it (C.G.); raffaella.rebucci@unimi.it (R.R.); antonella.baldi@unimi.it (A.B.); 2Department of Microbiology and Immunology, University of Veterinary Medicine and Pharmacy in Košice, Komenského 68/73, 04181 Košice, Slovakia

**Keywords:** omega-3 polyunsaturated fatty acids, eicosapentaenoic acid, docosahexaenoic acid, porcine IPEC-J2 cells

## Abstract

**Simple Summary:**

Farm animals frequently suffer from chronic inflammatory diseases due to certain physiological or pathophysiological conditions such as weaning, the periparturient period and infections. Traditionally, antibiotics were added to animal diets to counteract inflammation and enhance growth. However, this leads to the emergence of antibiotic-resistant bacterial species which causes potential health hazards. Over several decades, omega-3 polyunsaturated fatty acids have been known to exhibit a multitude of beneficial effects in animal health and are regarded as a functional food with therapeutic potential. We accessed the bioactivity of omega-3 polyunsaturated fatty acids as eicosapentaenoic acid and docosahexaenoic acid in pig intestinal epithelium under different stress conditions in an in vitro set-up. Our results demonstrated the proliferative and cytoprotective properties of the two fatty acids, which are fundamental to determining the cellular mechanism for efficient utilization in pig diets.

**Abstract:**

Marine and plant-based omega-3 polyunsaturated fatty acids (ω-3 PUFAs) are widely added to animal diets to promote growth and immunity. We tested the hypothesis that eicosapentaenoic acid (EPA), docosahexaenoic acid (DHA) and their 1:2 combination could counteract acute or long-term damage of lipopolysaccharides (LPS), dextran sodium sulphate (DSS) and hydrogen peroxide (H_2_O_2_) in Intestinal Porcine Epithelial Cell line-J2 (IPEC-J2). The results showed that 24 h treatment with EPA or DHA exhibited proliferative effects in IPEC-J2 cells at low to moderate concentrations (6.25–50 μM) (*p* < 0.05). Further, 24 h pretreatment with individual DHA (3.3 µM), EPA (6.7 µM) or as DHA:EPA (1:2; 10 µM) combination increased the mitochondrial activity or cell membrane integrity post-LPS (24 h), DSS (24 h) and H_2_O_2_ (1 h) challenge (*p* < 0.05). Additionally, DHA:EPA (1:2, 10 µM) combination decreased the apoptotic caspase-3/7 activity around twofold after 24 h LPS and DSS challenge (*p* < 0.05). Our study confirms the proliferative and cytoprotective properties of EPA and DHA in IPEC-J2 cells. Increased intracellular mitochondrial activity and cell membrane integrity by ω-3 PUFAs can play a role in preventing enterocyte apoptosis during acute or chronic inflammatory and oxidative stress.

## 1. Introduction

The intestinal epithelial layer (IEL) at the gut–lumen and tissue interface is an important key player of host–innate immunity. The IEL maintains intestinal homeostasis by establishing communication between gut microbiota and underlying immune cells. Further, it acts as a “physical barrier” that blocks the entry of luminal pathogens and antigens into the circulatory system [1]. Sometimes, excessive inflammation and oxidative stress can damage the integrity of the IEL and contribute to intestinal malfunctioning. Pigs during weaning in particular undergo severe inflammatory and metabolic stress due to altered diet, separation from piglets, transportation and exposure to pathogens. Further, the weaning process alters the physiological structure and functions of the intestine which is subsequently predisposed to growth deficiencies, morbidity and mortality [2]. Dietary omega-3 polyunsaturated fatty acids (ω-3 PUFAs) such as eicosapentaenoic acid (EPA) and docosahexaenoic acid (DHA) are widely known to exhibit strong anti-inflammatory, anti-oxidative and other health-beneficial properties in animals [3,4,5]. Generally, ω-3/-6 PUFAs are structural constituents of phospholipid cell membranes which maintain cell flexibility and fluidity. They also perform immunomodulatory functions by generating lipid-based signaling mediators. It is widely assumed that ω-6 derived mediators play a central role in pro-inflammation and the development of inflammatory diseases. Recently, however, it has been found that ω-3 can replace ω-6 in immune cell membranes and give rise to low potency lipid-mediators that act as agonists of pro-inflammation [5,6,7]. These principal findings are supported by numerous animal studies which reported that dietary supplementation of ω-3 PUFAs ameliorates lipopolysaccharides (LPS) mediated inflammation and weaning stress and improves intestinal structure during gestation and lactation in pigs [8,9,10,11]. Previously, its anti-inflammatory potential was tested in cancer enterocyte models [12,13]. However, cancer cells are not a realistic representative of normal physiology [14,15,16]. Recently, ω-3 PUFAs were shown to reverse mycotoxin damage of non-transformed IPEC-1 cells [17]. However, there has been limited research investigating the ω-3 PUFA effects on IEL injury in various stress conditions. Here, the ability of ω-3 PUFAs to counteract intracellular and membrane damage mediated by different stressors as LPS, dextran sodium sulphate (DSS) and hydrogen peroxide (H_2_O_2_) was evaluated in a non-transformed porcine enterocyte model, IPEC-J2.

## 2. Materials and Methods

### 2.1. Cell Line and Culture Conditions

IPEC-J2 is a non-transformed cell line, derived from the jejunum epithelium of unsuckled piglet (DSMZ, Braunschweig, Germany). Cell passages of 24–28 were used for the experiments. Cells were cultured in a complete medium consisting of a 1:1 mixture of Dulbecco’s modified Eagle’s medium with stable L-Glutamate and Ham’s F-12 mixture (DMEM/F12) (Immunological sciences, Società Italiana Chimici, Rome, Italy), supplemented with 15 mM HEPES (Sigma-Aldrich, Milano, Italy), 5% heat-inactivated fetal bovine serum (FBS) (Immunological sciences, Società Italiana Chimici, Rome, IT) and 1% penicillin (100 U/mL)/streptomycin (100 mg/mL) (Euroclone, Milano, Italy). Cells were seeded at a density of 2 × 10^6^ cells per T-75 cm^2^ culture flask (Corning, Sigma-Aldrich, Milano, Italy) and maintained at 37 °C under a humidified atmosphere with 5% CO_2_. At complete confluency, cells were trypsinated with 5 mL trypsin-EDTA (Sigma-Aldrich, Milano, Italy). Subsequently, the viable cells were counted in a hemocytometer by tryphan blue (Sigma-Aldrich, Milano, Italy) staining and seeded at a density of 3 × 10^4^ cells/well in 96-well plates (Corning, Sigma-Aldrich, Milano, Italy). Cells were allowed to adhere 24 h prior to the treatments.

### 2.2. Cell Treatments

#### 2.2.1. Dose-Response Study

Cells were treated with increasing concentrations of EPA or DHA (0–200 µM) in DMEM medium for 24 h. Similarly, the cells were exposed to different stressors such as *Salmonella typhimurium* LPS (0–100 µg/mL), DSS (0–10%) for 24 h and H_2_O_2_ (0–10 mM) for 1 h. All chemicals were purchased from Sigma-Aldrich (Milano, Italy).

#### 2.2.2. Cellular Challenge

Cells were pre-treated for 24 h with or without DHA (3.3 µM), EPA (6.7 µM) and DHA:EPA (1:2; 10 µM) in a DMEM medium, supplemented with 0.05% FBS. The FBS concentration was chosen based on a preliminary study with a range of 0–1% up to 48 h. The concentration of 0.05% FBS optimally supported cell attachment and growth. Subsequently, the cells were challenged with LPS (50 µg/mL), DSS (2%) for 24 h and H_2_O_2_ (1 mM) for 1 h individually to induce inflammatory and oxidative damage.

### 2.3. Cell Viability Assay

Cell viability was determined by quantification of mitochondrial oxidoreductase using the (3-(4,5-dimethylthiazol-2-yl)-2,5-diphenyltetrazolium bromide (MTT) assay method. Following cellular treatments (1.2b), the supernatant was replaced with 150 µL of MTT buffer (Sigma-Aldrich, Milano, Italy) prepared in 1× PBS (0.25 mg/mL) for 2 h. The end product so obtained was dissolved in an equal volume of dimethyl sulfoxide (DMSO) (Sigma-Aldrich), and the optical density (OD) was measured at 570 nm in a calorimetric plate reader (Bio-Rad, Sigma-Aldrich). Cells without any treatment were included as a control which represented 100% viability and DMSO was used as blank. Cell viability was calculated using the formula:Cell viability (%) = (OD_treatment_ − OD_blank_/(OD_control_ − OD_blank_) × 100(1)

### 2.4. Lactate Dehydrogenase Assay

Cell membrane integrity was estimated by assessing the cytosolic lactate dehydrogenase (LDH) released in the culture media. After the treatments (section 1.2b), 50 µL of the supernatant was mixed with an equal volume of LDH buffer (CytoTox 96^®^, Promega, Madison, WI, USA) in a 96-well plate and incubated for 30 min in darkness, at room temperature. Subsequently, colorimetric measurement was performed in a microplate reader (Bio-Rad) at 490 nm.

### 2.5. Nitric Oxide Activity

The amount of cellular nitric oxide (NO) in the culture media was estimated via the Griess test. For this, 50 µL of supernatant from the treatments (section 1.2b) was incubated with Griess reagent (Promega). The end-point was colorimetrically measured at 490 nm as per the manufacturer’s instructions.

### 2.6. Apoptosis Assay

Cellular apoptosis was determined by accessing the intracellular caspase-3/7 activity. For this, the cells were seeded at a low density of 2 × 10^4^ cells/well in a 96-well plate and the treatments were performed as described above (section 1.2b). Thereafter, supernatant was replaced with assay reagent, incubated in darkness and fluorescence was measured at 499_Ex_/521_Em_ in a microplate reader (Bio Tek, Agilent, Milano, Italy) as per manufacturer’s instructions (Apo-ONE^®^, Promega).

### 2.7. Statistical Analysis

The values of half inhibitory concentrations (IC_50_) were calculated by non-linear regression analysis using GraphPad Prism software (Version 8). Treatment effects were assessed by one-way ANOVA with Bonferroni post-hoc test. At least three independent experiments were performed with three technical replicates. All data are expressed as mean ± SEM and a value of *p* < 0.05 was considered statistically significant.

## 3. Results

### 3.1. Dose-Response Effect of Different Test Compounds on the IPEC-J2 Cell Viability

As shown in Figure 1a,b, the concentrations between 6.25 and 50 μM of EPA and DHA significantly increased cell viability which abruptly decreased at 200 µM compared to the control (0 μM, 100% viability; *p* < 0.05). The dose–response curves of LPS, DSS and H_2_O_2_ are reported in Figure 2a–c. Twenty-four hours exposure to LPS did not affect IPEC-J2 cell viability until 25 μg/mL (Figure 2a) and DSS until 0.62% (Figure 2b). Later, it decreased in a dose-dependent manner between 50 and100 μg/mL for LPS and 1.25% and 10% for DSS. A similar dose-dependent effect was observed for 1 h acute challenge of H_2_O_2_ as shown in Figure 2c. Subsequently, the dose–response curves of the stressors were used to estimate the IC_50_ values (section 1.7) that correspond to 62.02 μg/mL for LPS, 2.07% for DSS and 1.53 mM H_2_O_2_, respectively. Based on these values, the concentrations of 50 µg/mL LPS, 2% DSS and 1 mM H_2_O_2_ below the IC_50_ were chosen for further nutrient assessment.

### 3.2. Viability Effects of EPA and DHA on IPEC-J2 Cells Challenged by Different Stressors

MTT-based viability assay was used to determine the treatment effects on mitochondrial metabolic activity of IPEC-J2 cells. Pre-treatment of cells with EPA (6.7 µM) and DHA:EPA (1:2; 10 µM) increased mitochondrial activity against LPS stress, while DHA did not have an impact compared to the control (0 µM, *p* < 0.05) (Figure 3a). Similar mitochondrial protection was observed for cells pre-treated with DHA (3.3 µM), EPA (6.7 µM) and their combination, DHA:EPA (1:2; 10 µM) from the damage of 24 h DSS (Figure 3b) and 1 h H_2_O_2_ stress (Figure 3c) compared to the control (0 µM, *p* < 0.05).

### 3.3. Effects of EPA and DHA on Membrane Integrity and NO Activity of IPEC-J2 Cells Challenged by Different Stressors

LDH assay was performed to determine the treatment effects on cell membrane integrity. Neither the individual DHA (3.3 µM)/EPA (6.7 µM) nor its combination DHA:EPA (1:2; 10 µM) were able to recover membrane integrity from 24 h LPS (Figure 4a) and DSS challenge (Figure 4b) (*p* < 0.05). As shown in Figure 4c, pre-treatment of the cells with DHA (3.3 µM), EPA (6.7 µM) and DHA:EPA (1:2; 10 µM) was able to significantly maintain membrane integrity against 1 h H_2_O_2_ challenge compared to the control (0 µM, *p* < 0.05)_._ The NO activity in the culture media was tested by Griess reagent. Surprisingly, the NO content in the culture media was below the detection limit (<2.5 µM) for all the treatments (not shown).

### 3.4. Effects of EPA and DHA on Apoptosis of IPEC-J2 Cells Exposed to Different Stressors

The intracellular caspase-3/7 activity was estimated to elucidate the treatment effects on apoptosis of IPEC-J2 cells. A twenty-four hours challenge of 50–100 µg/mL LPS (Figure 5a) and 2% DSS (Figure 5b) significantly increased the caspase-3/7 activity twofold compared to the control (0 µM) while, the 24 h DHA:EPA (1:2; 10 µM) pre-treatment remarkably reduced this activity in IPEC-J2 cells (*p* < 0.05). Additionally, the reduction of caspase-3/7 activity was comparatively higher for 50 µg/mL over 100 µg/mL of LPS. In contrast, no caspase-3/7 activity was observed for H_2_O_2_ challenge (not shown).

## 4. Discussion

Supplementation with ω-3 PUFAs of human and animal diets has been reported to control a wide range of inflammatory diseases [18,19,20,21]. It also improves reproductive performance in farm animals [22,23,24]. Nevertheless, its molecular mechanism at the level of the IEL under inflammatory and oxidative stress remains uncertain. It is therefore essential to delineate the effects of ω-3 PUFAs using appropriate cell models for efficient utilization in animal diets. Previously, the significance of cell models and cell-based assays available for screening antioxidants was described [25]. In particular, intestinal cell models are widely utilized for the validation of bioactive compounds in food and feed analysis [15,26,27,28]. Earlier, the anti-inflammatory and anti-necroptosis activity of ω3-PUFAs were demonstrated in normal and cancer enterocyte models [12,13,17]. However, cancer cells face certain limitations as they exhibit aberrant cell surface sugars, glycoproteins, glycosylation and metabolic pathways [14,15,16].

Here, we demonstrated the cytoprotective properties of ω-3 PUFAs (EPA, DHA, DHA:EPA) in the porcine IPEC-J2 enterocyte model under different biological and chemical stress. IPEC-J2 is a non-cancer and non-transformed enterocyte model. Moreover, it is isolated from the jejunum epithelium, the primary site of nutrients absorption. Furthermore, enterocytes are the major cell population of IEL (~70–80%), making it an appropriate in vitro cell model to delineate nutrient behavior closer to an in vivo environment.

The pathophysiological events of inflammation and oxidative stress were stimulated in the IPEC-J2 cells by challenge against LPS, H_2_O_2_ and DSS. LPS is a component of the Gram-negative bacterial cell wall which instigates pro-inflammatory responses by activation of the NF-κB signalling pathway [29]. In addition, LPS triggers oxidative stress by decreasing the Nrf2 transcription factor that regulates cellular redox balance. H_2_O_2_ is a commonly used chemical in IPEC-J2 cells for the characterization of dietary antioxidants [30,31,32]. It induces oxidative stress-mediated inflammation by altering the subcellular DNA, proteins and lipids [33]. DSS is mainly used in animal models to induce a leaky epithelial barrier for the stimulation of inflammatory bowel disease. This is the first report of DSS application in IPEC-J2 cells, and earlier, it was used in cancer cells [34].

Further, it is important to apply an optimal concentration of the stressor that renders mild toxicity without completely killing or causing irrecoverable cell damage. To estimate this, a dose–response study was performed by exposing the cells to increasing concentrations of each stressor. Their corresponding IC_50_ values were then determined from the viability assay. Subsequently, the stressor concentrations for ω-3 PUFA assessment (section 1.2b) were chosen below the IC_50_ range.

We observed that LPS concentrations > 25 µg/mL are required to decrease the viability of IPEC-J2 cells after 24 h exposure. It was reported that high concentrations of LPS between 100 and 300 µg/mL are required to reduce the viability of non-transformed rat IEC-18 enterocytes after 24 h [35]. LPS concentrations between 1 and 10 µg/mL did not affect IPEC-J2 viability until 24 h, while 10 µg/mL activated the expression of pro-inflammatory IL6 and IL8 mRNA [36]. An LPS concentration of 10 µg/mL activated pro-inflammatory genes in 8 h, although it did not cause any cytotoxic effects after 24 h [37]. From these observations, in addition to the outcome of our present dose–response study, it is evident that LPS concentrations between 0.1 and 10 µg/mL (1–24 h) are sufficient to activate pro-inflammatory genes, while concentrations > 25 µg/mL (24 h) are required to induce cytotoxic effects in IPEC-J2 cells. Thus, an LPS concentration of 50 µg/mL (24 h) below the IC_50_ range was chosen to investigate the ω-3 PUFA effects against biological damage. Likewise, the concentration of 1 mM H_2_O_2_ (1 h) and 2 % DSS (24 h) was chosen to evaluate the ω-3 PUFA effects against chemical damage. Similar concentrations of H_2_O_2_ were previously reported to reduce IPEC-J2 viability [30,31,32,33]. Additionally, 2% DSS was used to induce inflammation in the cancer cell models [34].

From the dose–response study of EPA and DHA, a low to moderate concentration between 6.25 and 50 μM was observed to significantly increase viability, while 200 μM exhibited inhibitory effects in IPEC-J2 cells. A similar increase in cell viability at 6.25–25 μg/mL of EPA and DHA was observed in IPEC-1 cells [17]. Further, EPA and DHA were reported to induce a dose-dependent anti-proliferative, anti-cancer and apoptosis activity at 15–200 μM in human cancer enterocytes as HT-29, Caco-2 and DLD-1 [38,39,40,41]. In another study, cancer cells were reported to exhibit differential ω-3 PUFA uptake and membrane distribution due to decreased activity of ω-3 PUFA synthesizing Δ^5^/Δ^6^ desaturase enzymes over normal cells [42,43,44]. In addition, excessive production of reactive oxygen species by cancer cells was reported to accelerate ω-3 PUFA mediated lipid peroxidation, making it more vulnerable to ω-3 PUFA than normal cells [44]. These factors could explain the differential outcome of ω-3 PUFA activity in cancer and non-cancer enterocytes. Therefore, it is essential to choose appropriate cell models to avoid misinterpretations on the outcome of a study. A concentration of 10 µM EPA/DHA was chosen as a starting point to determine its lowest cytoprotective dose in IPEC-J2 cells. Furthermore, 10 µM was the minimal concentration reported to significantly exhibit anti-inflammation in macrophages [45] and 12.5–25 μg/mL in IPEC-1 cells [17]. Additionally, 10 µM lies in the range of concentrations that exhibited proliferative effects in our dose–response study. Normally, the natural sources of ω-3 PUFA as fish and linseed oil comprise a mixture of short and long chain fatty acids with major proportions of EPA and DHA in different ratios. In order to mimic this natural co-existence, we mainly focused on the effects of DHA:EPA combination in a 1:2 ratio as chosen from previous in vivo trials [46,47,48]. Our experimental set-up enabled us to demonstrate that the pre-treatment of IPEC-J2 cells with DHA (3.3 µM), EPA (6.7 µM) and DHA:EPA (1:2; 10 µM) is able to significantly restore the mitochondrial activity reduced by H_2_O_2_ (1 h) and DSS (24 h) induced stress as observed from the viability assay. Similarly, EPA (6.7 µM) and DHA:EPA (1:2; 10 µM) protected the mitochondrial activity against LPS (24 h) stress while single DHA (3.3 µM) administration did not show any impact. Possibly, its concentration is too low to confer cytoprotection unlike EPA or the combination.

Moreover, as witnessed from the LDH assay, DHA (3.3 µM), EPA (6.7 µM) and DHA:EPA (1:2; 10 µM) potentially recovered membrane integrity from H_2_O_2_ stress, while they were unable to do so for LPS and DSS stress. The concentrations of LPS and DSS tested in the current study are speculated to have disrupted the IPEC-J2 cell membrane beyond the recovery potential of different nutrient doses tested. Additionally, post-LPS and DSS challenge, a two-fold remarkable decline in apoptotic caspase-3/7 was observed for cells with DHA:EPA (1:2; 10µM) pre-treatment than for those without. In contrast, no caspase-3/7 activity was observed for H_2_O_2_. Perhaps the acute challenge of H_2_O_2_ activated necrosis instead of apoptosis, which requires further analysis. Generally, cellular apoptosis is activated by multiple pathways such as DNA fragmentation, certain mitochondrial proteins, or electrolyte imbalance, induced by membrane damage [49]. Thus, the anti-apoptotic activity of ω-3 PUFA observed in IPEC-J2 cells could be mediated by control of mitochondrial activity and membrane integrity. In accordance with our results, ω-3 PUFA was shown to recover mitochondrial activity, membrane integrity and suppressed necroptosis genes corresponding to mycotoxin injury in IPEC-1 cells [17]. Future investigation is necessary to access gene and protein expressions to substantiate the pathway involved in ω-3 PUFA bioactivity. This study has a practical application of using ω-3 PUFA as a nutritional strategy in pigs. Thus, the present outcome is fundamental to understanding the cellular mechanism underpinning ω-3 PUFA nutrition in pig diets.

## 5. Conclusions

To summarize, our current study demonstrated the proliferative and cytoprotective properties of ω-3 PUFAs as EPA and DHA in the non-transformed porcine enterocyte model, IPEC-J2. EPA and DHA, either individually or in combination, are able to suppress the enterocyte apoptosis through recovery of mitochondrial activity or cell membrane integrity from the detrimental effects of LPS, DSS and H_2_O_2_. Additionally, for the first time, low concentrations of ω-3 PUFAs were demonstrated to efficiently control both acute and long-term damage. Further, these in vitro effects should be translated into in vivo efficacy through animal trials for dose–effect optimization.

## Figures and Tables

**Figure 1 animals-10-00956-f001:**
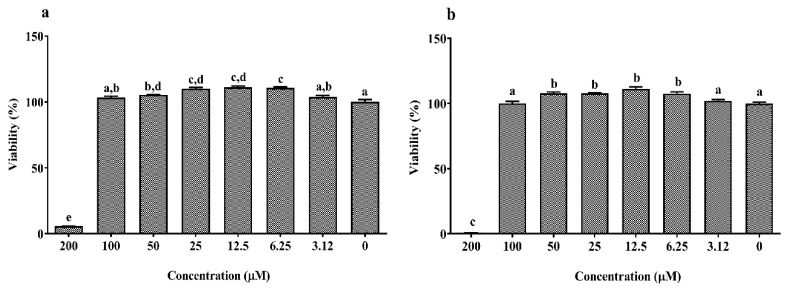
Dose–response effect of (**a**) eicosapentaenoic acid (EPA) and (**b**) docosahexaenoic acid (DHA) with different concentrations (0–200 µM) on the Intestinal Porcine Epithelial Cell line (IPEC-J2) viability for 24 h. Data are presented as mean ± SEM (*n* = 3, one-way ANOVA). Statistically significant differences among the treatments are denoted by different letters (a, b, c, d, e), where *p* < 0.05.

**Figure 2 animals-10-00956-f002:**
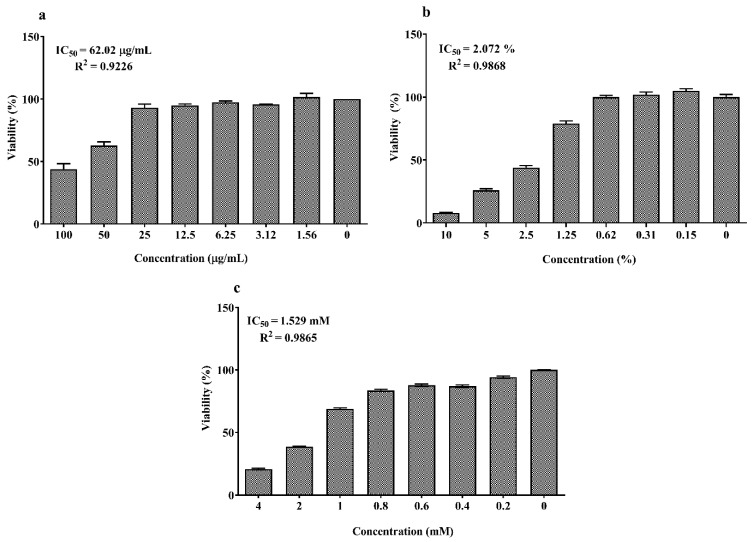
Dose–response effect of (**a**) lipopolysaccharide (LPS) (0–200 µM) for 24 h; (**b**) dextran sodium sulfate (DSS) (0–10%) for 24 h and (**c**) H_2_O_2_ (0–4 mM) for 1 h on IPEC-J2 viability. Data are presented as mean ± SEM (*n* = 3, one-way ANOVA). The 50% inhibitory concentrations (IC_50_) were calculated by non-linear regression analysis and R^2^ represents the goodness of fit.

**Figure 3 animals-10-00956-f003:**
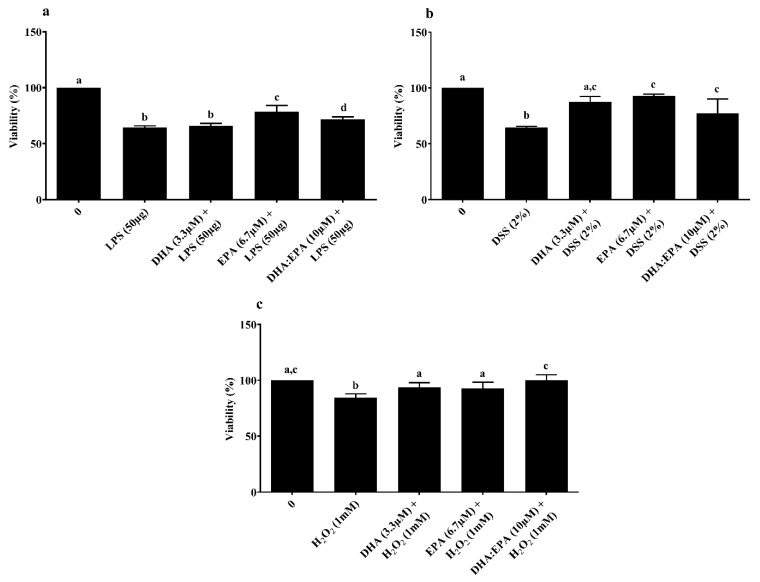
Effects of DHA (3.3 µM), EPA (6.7 µM) and DHA:EPA (1:2; 10 µM) pre-treatment (24 h) on the viability of IPEC-J2 cells challenged by (**a**) LPS (50 or 100 µg/mL) for 24 h; (**b**) DSS (2%) for 24 h and (**c**) H_2_O_2_ (1 mM) for 1 h. Data are presented as mean ± SEM (*n* = 3, one-way ANOVA). Statistically significant differences among the treatments are denoted by different letters (a, b, c, d), where *p* < 0.05.

**Figure 4 animals-10-00956-f004:**
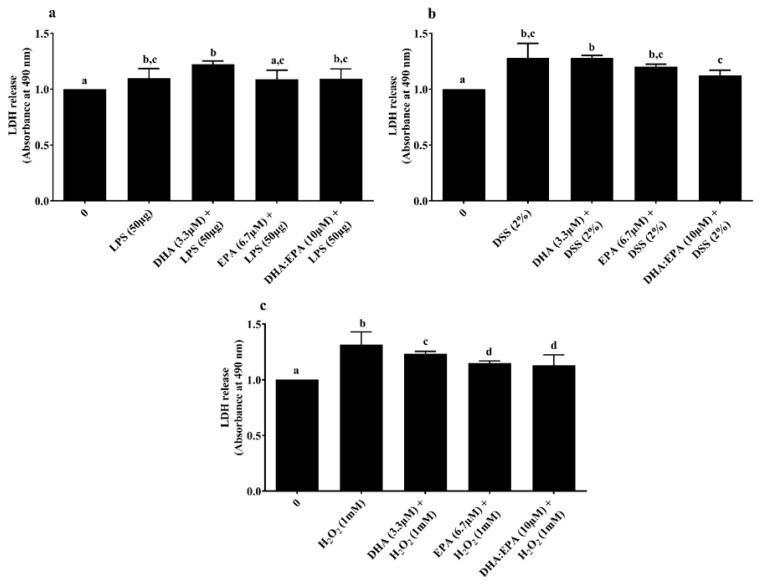
Effects of DHA (3.3 µM), EPA (6.7 µM) and DHA:EPA (1:2; 10 µM) pre-treatment (24 h) on the membrane integrity of IPEC-J2 cells challenged by (**a**) LPS (50 or 100 µg/mL) for 24 h; (**b**) DSS (2%) for 24 h and (**c**) H_2_O_2_ (1 mM) for 1 h. Data are presented as mean ± SEM (*n* = 3, one-way ANOVA). Statistically significant differences among the treatments are denoted by different letters (a, b, c, d), where *p* < 0.05.

**Figure 5 animals-10-00956-f005:**
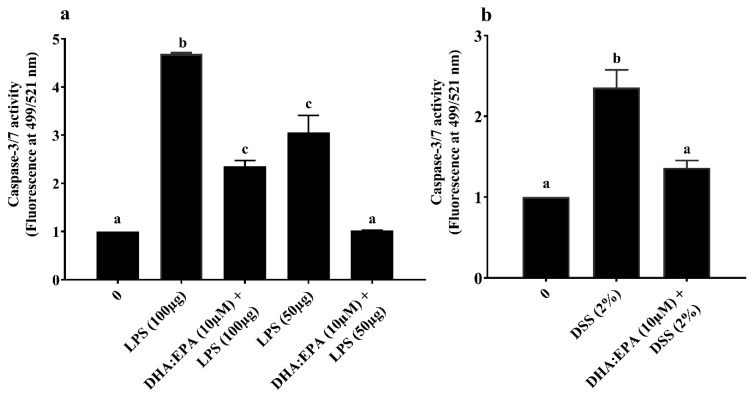
Effects of DHA:EPA (1:2; 10 µM) pre-treatment (24 h) on the apoptosis of IPEC-J2 cells challenged by (**a**) LPS (50 or 100 µg/mL) and (**b**) DSS (2%) for 24 h. Data are presented as mean ± SEM (*n* = 3, one-way ANOVA). Statistically significant differences among the treatments are denoted by different letters (a, b, c), where *p* < 0.05.

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
