# Peer review of "Omega-3 Polyunsaturated Fatty Acids Counteract Inflammatory and Oxidative Damage of Non-Transformed Porcine Enterocytes"

_animals, 2020, doi:10.3390/ani10060956_

Round 1

Reviewer 1 Report

During the review process, some major concerns were raised.

  1. Line 133, the dose-response effect suggested that the concentrations between 6.25-50 μM of EPA and DHA, significantly increased the cell viability. Then, why did you use 3.3 μM of DHA in the following experiments?
  2. Why did you choose the concentrations of 3.3 μM of DHA and 6.7 μM of EPA but not other concentrations to treat cells? And why did you choose the ratio of DHA to EPA at 2:1?
  3. The different effects of DHA, EPA, and their combination on cells could be a result of different doses (3.3, 6.7 and 10 μM ), but not different PUFA.
  4. I did not see any results related to mitochondrial activity. How did you get the conclusion that PUFA increased mitochondrial activity.
  5. To make the results and conclusion more convincible, please add the results of apoptosis by TUNEL analysis and present representative pictures.
  6. SEM were missing in control groups in Figure 3, 4, 5. n number should be added in all figure legend.
  7. line 155, figure 3b, figure 3c is mentioned before Figure 3a in text. The sequence should be adjusted.
  8. line 168, figure 4c is mentioned before Figure 4a and 4b in text. The sequence should be adjusted.
  9. line 207-209, “we demonstrated the cytoprotective properties of ω-3 PUFAs as EPA and DHA that potentially counteracted both inflammatory  and  oxidative damage  of  porcine  IPEC-J2 enterocyte model.”  The authors did not present any results related to inflammatory  and  oxidative responses. How did you get this conclusion?

10. In figure 4a, explain why 3.3 μM of DHA exacerbated the LDH release?

Reviewer 2 Report

Dear Authors,

The present manuscript is important and actual work. The manuscript analyzed the literature works in detail and at high level of discussion. The only my recommendation - to check the reference’s style and consider those in the rather local Journals (for example, references 32, 42).

I do not doubt the technical quality of the work and feel that there is a sufficient impact on a broader readership to justify publication in the "Animal". This topic is in frame of the journal scope, the subject matter is treated in depth.

The manuscript is devoted to the study of the  “ability of ω-3 PUFAs in counteracting the inflammatory and oxidative damage mediated by different  stressors as bacterial LPS, DSS and H2O2 was evaluated in a non-transformed porcine enterocyte   model, IPEC-J2”.  The authors found very interesting facts: “the proliferative and cytoprotective properties  of ω-3 PUFAs as EPA and DHA in the non-transformed porcine enterocyte model, IPEC-J2.. ”. Moreover, the “EPA  and DHA, either individually or in combination are able to suppress the enterocyte apoptosis  through recovery of mitochondrial activity or cell membrane integrity from the inflammatory and oxidative stress. Additionally, for the first time, low concentrations of ω-3 PUFAs were  demonstrated to efficiently control both acute and long-term damage. Further, these in vitro effects should be translated into in vivo efficacy through animal trials for dose-effect optimization”.

I consider the manuscript for publication with small changes (minor revision) as follows:

  1. to check the reference’s style in the reference list .

1.1. It is important to add the Journal name (Cancer Res.) in the ref.14 (Hakomori, S.-i., Tumor Malignancy Defined by Aberrant Glycosylation and Sphingo(glyco)lipid 327 Metabolism. 1996, 56 (23), 5309-5318 ).

1.2. It is important to add the Journal name (J. Immunol.) in the ref.28 (Dziarski, R.; Wang, Q.; Miyake, K.; Kirschning, C. J.; Gupta, D., MD-2 Enables Toll-Like Receptor 2 362 (TLR2)-Mediated Responses to Lipopolysaccharide and Enhances TLR2-Mediated Responses to 363 Gram-Positive and Gram-Negative Bacteria and Their Cell Wall Components. 2001, 166 (3), 1938-1944).  

  1. I propose to add appropriate references, rewrite or cut some general sentences.

2.1. For example, in the part 1 (Introduction):  the sentence on the Page 2, lines 48-50: “Thus, the integrity of IEL is of paramount importance in the maintenance of healthy gut.  Sometimes, excessive inflammation and oxidative stress can damage the IEL and contribute to intestinal malfunctioning. As a consequence, animals suffer from poor nutrition, dysentery, loss of  body weight as well as chronic and metabolic disorders”. For example, in the part 4 (Discussion):

2.2. The  first sentence after “Discussion” (Page 8, lines 192-195): “Certain physiological and pathophysiological conditions as weaning, periparturient and  infection, develops extraneous inflammation and oxidative stress in animals. As a consequence, the integrity of IEL is detrimentally affected, resulting in intestinal malfunction. Furthermore,  inflammation and oxidative stress induces poor nutrition and growth performance as a large part of the dietary energy is spent in the maintenance of homeostasis”.

2.3. The first sentence (Page 10, lines 268-270): “Generally, cellular apoptosis is activated by multiple pathways such as DNA fragmentation, electrolyte imbalance created during membrane damage or through certain proteins released by  mitochondria”.  

Reviewer 3 Report

General comment

This is a highly relevant work on the protective effects of omega-3-fatty acids on enterocytes.

Your manuscript needs language editing.

Can you please further explain the toxicity at high concentrations? This should be discussed in more detail and you should comment in more detail on the different outcomes in cancer and non-cancer enterocytes. Thank you.

Specific suggestions for minor changes

Line 17: fatty acids are known

Line 18: and are regarded

Line 24: acids are widely

Line 29: by challenging them with the three different stressors lipopolysaccherides

Line 48: of a healthy gut

Line 236: that LPS concentrations ... are sufficient

Line 237: concentrations ... are required

Line 244: a low to moderate concentration

Line 245: , while high concentrations

Line 246: EPA and DHA exhibit

Line 256: to significantly restore mitochondrial activity reduced by ... induced stress

Line 259: against LPS

Line 259: while, single DHA administration

Round 2

Reviewer 1 Report

I have no further comments